# Prognostic Factors for Restenosis of Superficial Femoral Artery after Endovascular Treatment

**DOI:** 10.3390/jcm12196343

**Published:** 2023-10-03

**Authors:** Vinko Boc, Matija Kozak, Barbara Eržen, Mojca Božič Mijovski, Anja Boc, Aleš Blinc

**Affiliations:** 1Department of Vascular Diseases, University Medical Centre Ljubljana, 1000 Ljubljana, Slovenia; matija.kozak@kclj.si (M.K.); barbara.erzen@kclj.si (B.E.); mojca.bozic@kclj.si (M.B.M.); anja.boc@mf.uni-lj.si (A.B.); ales.blinc@kclj.si (A.B.); 2Faculty of Medicine, Department of Internal Medicine, University of Ljubljana, 1000 Ljubljana, Slovenia; 3Faculty of Pharmacy, University of Ljubljana, 1000 Ljubljana, Slovenia; 4Faculty of Medicine, Institute of Anatomy, University of Ljubljana, 1000 Ljubljana, Slovenia

**Keywords:** endovascular procedures, femoral artery, vascular patency, hemostasis, polymorphisms

## Abstract

High incidence of superficial femoral artery (SFA) restenosis after percutaneous transluminal angioplasty (PTA) poses a persistent challenge in peripheral arterial disease (PAD) treatment. We studied how the patients‘ and lesions’ characteristics, thrombin generation, overall haemostatic potential (OHP), and single nucleotide polymorphisms (SNPs) of the NR4A2 and PECAM1 genes affected the likelihood of restenosis. In total, 206 consecutive PAD patients with limiting intermittent claudication due to SFA stenosis who were treated with balloon angioplasty with bailout stenting when necessary were included. Patients’ clinical status and patency of the treated arterial segment on ultrasound examination were assessed 1, 6, and 12 months after the procedure. Restenosis occurred in 45% of patients, with less than 20% of all patients experiencing symptoms. In the multivariate analysis, predictors of restenosis proved to be poor infrapopliteal runoff, higher lesion complexity, absence of treated arterial hypertension, delayed lag phase in thrombin generation, and higher contribution of plasma extracellular vesicles to thrombin concentration. Poor infrapopliteal runoff increased the risk of restenosis in the first 6 months, but not later. The negative effect of poor infrapopliteal runoff on SFA patency opens questions about the potential benefits of simultaneous revascularisation of below-knee arteries along with SFA revascularisation.

## 1. Introduction

In selected patients with peripheral arterial disease (PAD), limb revascularisation is needed to either improve quality of life or to prevent or at least minimize tissue loss. Endovascular procedures are nowadays the generally accepted first-line revascularisation method [1,2]; however, the incidence of superficial femoral artery (SFA) restenosis after percutaneous transluminal angioplasty (PTA) remains high despite the progress in both procedure technology and accompanying pharmacologic therapy [3,4]. Consequently, factors leading to SFA restenosis are continuously being studied. Among well-established risk factors are the clinical presentation of PAD, the morphological characteristics of arterial lesions, and the presence of certain risk factors for atherosclerosis, such as diabetes mellitus and chronic renal failure [5,6,7,8]. Inadequate infrapopliteal runoff is also considered a risk factor for poor treatment outcomes, but data on the duration of its impact vary; while older studies observed a negative impact of poor outflow to last several years after PTA [7,9], a newer study confirmed the negative effect only during the first month after the revascularisation procedure, but not later [5,10]. The possibility of restenosis might also be influenced by haemostatic and genetic factors, but the knowledge about their impact is scarce. The aim of this study was to evaluate how the effect of poor infrapopliteal runoff, presence of risk factors for atherosclerosis, complexity of arterial lesions, haemostatic potential, and single nucleotide polymorphisms (SNPs) in the NR4A2 and PECAM1 genes affected the likelihood of SFA restenosis after balloon angioplasty with bailout stenting when necessary, and without the use of paclitaxel-coated devices.

## 2. Materials and Methods

This observational prospective single-centre study was performed in the Catheterisation Laboratory of the Department of Vascular Diseases, University Medical Centre Ljubljana, Slovenia, and included consecutive PAD patients with limiting intermittent claudication who underwent successful PTA of the native SFA between July 2010 and June 2015. Exclusion criteria were the presence of chronic limb-threatening ischemia (CLTI), use of a paclitaxel coated device, and long-term anticoagulant treatment due to its effect on the patients’ haemostatic properties.

In all patients, ankle-brachial pressure index (ABI) was recorded before PTA and again one day after the procedure. Values of ABI ≥ 1.4 were considered as mediocalcinosis and were not included in the statistical analysis. Risk factors for atherosclerosis were registered, and the pharmacologic therapy for their control was optimized according to the guidelines [11] in all patients with suboptimal treatment regimes.

Lesions were primarily treated with balloon dilatation, and bailout stenting was performed only in cases of significant residual stenosis, early elastic recoil, or flow-limiting artery wall dissection. In case of compromising lesions in popliteal and/or below-knee arteries, these lesions were also treated. After the procedure, patients with isolated SFA lesions treated solely with balloon dilatation were prescribed a single antiplatelet treatment with aspirin, while those with stent implantation or simultaneous intervention in the popliteal and/or below-knee arteries were prescribed transient dual antiplatelet therapy (DAPT), with the addition of clopidogrel to aspirin for three months. Interventions were performed by three different interventionists in one endovascular suite.

During the intervention, a pre-revascularisation angiogram was recorded to assess the complexity of lesion for Trans-Atlantic Inter-Society Consensus Document on Management of Peripheral Arterial Disease II (TASC II) classification [6]. Presence of SFA wall calcification was assessed by fluoroscopy. After revascularisation, a control angiogram was taken to evaluate the blood flow through the treated SFA segment and through the below-knee arteries. Infrapopliteal runoff was assessed by the modified Society for Vascular Surgery criteria, originally intended for quantifying bypass runoff, in which a higher score implies worse runoff [12]. According to the calculated outflow estimate, patients were divided into two groups: those with the sum < 5 (patent popliteal artery and at least two below-knee arteries without >50% narrowing) were included in the group with adequate runoff, while all other patients were included in the group with poor runoff.

One hour before the procedure, a venous blood sample was withdrawn from the median cubital vein and collected into vacuum tubes containing 0.11 mol/L sodium citrate (Becton Dickinson, Vacutainer System Europe, Germany). Plasma was prepared with 20-min centrifugation at 2000× *g*, aliquoted into plastic vials, snap-frozen in liquid nitrogen, and stored at −70 °C until analysis. In plasma, coagulation parameters D-dimer (Innovance D-dimer) and fibrinogen concentration (Dade^®^ Thrombin Reagent), prothrombin time (Thromborel S), and activated partial thromboplastin time (Pathromtin SL, all Siemens, Marburg, Germany) were measured on an automated coagulation analyser CS-2500 (Sysmex, Kobe, Japan). Thrombin generation and contribution of the plasma extracellular vesicles (EV) to thrombin generation were assessed with the Technothrombin^®^ TGA (Technoclone, Wien, Austria). Overall haemostatic potential (OHP) was determined as described previously [13]. In addition to OHP, overall coagulation potential (OCP) and overall fibrinolytic potential (OFP) were also reported. C-reactive protein (CRP) levels were determined with the Luminex Human Magnetic Assay (R&D Systems, ZDA, Minneapolis, MN, USA) on a MagPix instrument (Luminex, ZDA, Austin, TX, USA). DNA was isolated from whole blood using standard procedures. Genotype frequencies of the NR4A2 (rs1466408, rs13428968, rs12803) and PECAM1 genes (rs668, rs12953) were investigated.

Patients were followed for one year. One, six and twelve months after the procedure, clinical examination of the treated limb was performed, and the patency of the treated SFA was assessed by doppler ultrasound examination (DUS) performed by one of four experienced physicians. Morphology and haemodynamic characteristics of treated SFA segment were assessed, and peak systolic velocity (PSV) ratio was calculated as PSV measured at the site of the treated lesion and PSV measured in normal vessel proximal to the lesion. PSV ratio > 2 was defined as restenosis > 50%, and absence of a Doppler flow signal was defined as reocclusion [14]. At the final DUS, patency of the below-knee arteries was also re-evaluated.

Statistical data processing was performed with the SPSS statistical package (SPSS Statistics 25.0.0, IBM, Armonk, NY, USA). Categorical variables were presented as a number of units and shares. Continuous variables were tested for normality with the Kolmogorov-Smirnov test. Normally distributed variables were described by their mean and standard deviation, and variables with asymmetric distribution by their median and range between the 1st and 3rd quartiles. The groups’ characteristics were compared with the Pearson’s χ2 test for categorical variables, while continuous variables were compared with the independent samples two-tailed *t*-test if normally distributed, two-sided Mann-Whitney U test if asymmetrically distributed and independent, and Wilcoxon signed rank test if asymmetrically distributed and dependent. The Kruskal-Wallis test was used to analyse variance. Risk factors for atherosclerosis, arteriographic characteristics of the lesion, infrapopliteal runoff, and laboratory indicators of thrombin formation before the intervention were tested as possible predictors for restenosis with univariate regression analysis. Variables with a *p*-value < 0.05 on univariate analysis were then assessed in a multivariate regression analysis model.

## 3. Results

In total, 206 patients aged 37 to 88 years were enrolled in this study. Their baseline characteristics and characteristics of the treated lesions are presented in Table 1. Bailout stenting was necessary in 41 (20%) patients. In 103 (50%) patients, only SFA segment was revascularised, while in the other half of patients, popliteal and/or below-knee arteries were also treated. Poor postprocedural infrapopliteal runoff was present in 26 (12.6%) patients. The mean postprocedural ABI was significantly higher in comparison to preprocedural ABI (0.89 ± 0.17 vs. 0.64 ± 0.21, *p* < 0.001). Two patients were excluded from ABI analysis due to mediocalcinosis.

One year after the procedure, data about SFA patency were available for 98% (202/206) of patients; three patients were lost to follow-up, and one died three months after the revascularisation due to an unknown reason. Significant restenosis developed in 45% (91/202) of patients and was symptomatic in 39.6% (36/91) of patients; complete occlusion was present in 19.8% (18/91) of patients and was symptomatic in 72.2% (13/18) of patients. Among symptomatic patients, thirty-three out of thirty-six (91.7%) expressed limiting intermittent claudication, while the remaining three patients (8.3%) developed CLTI. One of the patients with CLTI underwent below-knee amputation without a prior revascularisation attempt, while all other symptomatic patients underwent successful endovascular reintervention.

A comparison of patients’ clinical features and characteristics of the treated lesion between the group of patients with restenosis and the group of patients without restenosis is presented in Table 2.

Restenosis was more frequent in patients with poor postprocedural runoff during the 1st month and also during the 2nd to 6th month after the procedure, but not later (Figure 1).

One year after the procedure, the infrapopliteal runoff was re-evaluated in 94.1% (190/202) of patients, in 160 patients with DUS, and in 30 patients angiographically during the reintervention due to symptomatic restenosis. Among all 190 patients, 168 had adequate runoff immediately after the procedure. In the subgroup of these 168 patients, restenosis was more frequent in those with deterioration of runoff compared to those with preserved adequate runoff (19% (12/63) vs. 6.6% (7/105), *p* = 0.014).

Table 3 contains the comparison of coagulation parameters, thrombin generation, indicators of haemostatic potential, and indicators of inflammation between the group of patients with restenosis and the group of patients without restenosis. We found statistically significant prolongation of the lag phase in thrombin generation, and a higher contribution of plasma extracellular vesicles (EV) to thrombin generation in patients with restenosis.

Genotyping of the prothrombotic polymorphisms in the NR4A2 and PECAM1 genes was performed in 160 patients. We found no difference in distribution of tested polymorphisms between the group of patients with restenosis and the group of patients without restenosis (Table 4). We also found no difference in the distribution of individual haplotypes of NR4A2 gene between the compared groups (Table 5).

Table 6 presents the results of multivariate analysis. Possible predictors of restenosis were poor infrapopliteal runoff, higher lesion complexity assessed by the TASC II, prolonged lag time in thrombin generation, larger contribution of plasma extracellular vesicles (EV) to thrombin generation, and the absence of treated arterial hypertension (Table 6).

## 4. Discussion

Revascularisation of SFA remains a challenging area in endovascular therapy due to poor long-term patency. This is especially true for patients with intermittent claudication, in whom the decision for vascular intervention is influenced by the threat of probable restenosis—opposite to the patients with CLTI, in whom the decision is more straightforward because of the limb-saving purpose of the procedure. We therefore explored poor infrapopliteal runoff, presence of risk factors for atherosclerosis, complexity of arterial lesions, haemostatic potential, and single nucleotide polymorphisms (SNPs) in the NR4A2 and PECAM1 genes as possible predictors for restenosis specifically in patients with intermittent claudication.

At the end of the one-year follow up period, significant restenosis confirmed with DUS was present in almost half of our patients, while symptomatic restenosis affected only 20% of all patients. These findings are comparable to those of other studies in which paclitaxel-eluting devices were not used, as in our study [9,15,16,17]. The literature provides limited insight into why some patients develop symptoms while others do not. Possible explanations include variations in collateral circulation development [18], differences in oxygen extraction efficiency from the blood [19], and altered gait biomechanics [20]. Collateral circulation permits even subjects with complete reocclusion of the treated artery to remain symptom-free; in our study, that was true in nearly 30% of cases. Physical capacity in patients does not always correlate with the extent of atherosclerotic changes in large arteries. Consequently, it is becoming increasingly common to evaluate the success of revascularisation intervention not only with image assessment of artery patency, but also with various functional tests [21]. However, in our study, we did not evaluate collateral circulation or subject the patients to functional tests to assess their physical performance.

Unexpectedly, SFA restenosis was less frequent in our patients with treated arterial hypertension in comparison to those without arterial hypertension, implicating its possible protective role. This discovery is supported by other studies as well [16,22]. While it could be an accidental finding, it is also possible that either the haemodynamic effect of high blood pressure on the revascularised artery wall or medications used for blood pressure lowering reduce the risk of restenosis. Unfortunately, we did not collect data on the actual blood pressure levels in our study participants during the follow-up period. As for therapy, the majority of our patients were receiving either angiotensin-converting enzyme inhibitors or angiotensin II receptor blockers, with no statistical difference between the group of patients with restenosis and the group of patients without restenosis.

Among our patients, ABI measured one day after the revascularisation was on average lower in the group of patients who developed restenosis compared to the group without restenosis. This finding is consistent with the research conducted by Gordon et al. [23]. The ABI value not only depends on the patency of the SFA, but also on the patency of the popliteal and below-knee arteries, which means it reflects the infrapopliteal runoff. Due to this possibility, we did not include ABI in the multivariate analysis.

In our study, restenosis was more than five times more likely in patients with poor postprocedural infrapopliteal runoff compared to those with adequate runoff. In concordance with our findings, a negative influence of poor runoff on the likelihood of restenosis was established in the studies conducted by Matsi et al. [8] and Davies et al. [9] that both included only patients with intermittent claudication, and in the study by Noh BG et al. [24] that explored patients with intermittent claudication treated with SFA stenting. On the contrary, Gordon et al. [23] did not confirm the impact of runoff on SFA restenosis in patients with intermittent claudication, but it is important to acknowledge that in their study which only encompassed 42 patients, the popliteal artery was not included in the calculation of infrapopliteal runoff. Similarly, Baril et al. [16] did not confirm the impact of the runoff on SFA patency in the mixed group of patients with either intermittent claudication or CLTI, counting the two patent tibial arteries as poor runoff. We also recognised the deterioration of initially adequate runoff as a predictive factor for restenosis. This finding was also observed in the study by Salapura et al. [10].

Of note, the impact of poor runoff on the SFA patency in our study was present during the first 6 months after the revascularisation, but not in the next 6-month period. Interestingly, Salapura et al. [10] confirmed the impact of poor outflow only in the first month after the intervention, and not later. They suggested that compromised infrapopliteal runoff could increase the risk of early SFA restenosis due to the reduced arterial blood flow predisposing treated arteries to thrombosis or early elastic recoil. In our study, the prolonged, yet still time-limited impact of the poor runoff could possibly be explained by more pronounced neointimal hyperplasia. That was also suggested by Hehrlein et al. [25] who found more extensive neointimal hyperplasia after balloon dilation of the femoral artery in dogs with poor outflow. Yamaguchi T et al. [26] also found an impact of runoff on the atherosclerotic burden in the femoropopliteal region, observing more ulcerated plaques in patients with poor runoff. Interestingly, Kaczmarczyk et al. [27] found a negative correlation between the blood flow velocity and the treatment outcome of endovascular procedures conducted on the infrainguinal arteries. This finding stemmed from the observation that patients with only one patent artery had higher blood flow velocity, whereas in patients with more than one patent artery, the velocity was comparatively lower due to distribution of the blood among patent vessels.

Since clinical presentation of PAD is linked to the complexity of atherosclerotic lesions [6,28], less than one quarter of TASC II C and D lesions in our patients with intermittent claudication is an expected share. Despite the low percentage of patients with complex lesions, results of our multivariate analysis of factors associated with restenosis demonstrate that lesion complexity remains a reliable predictor of long-term patency. Other studies that also indicated the negative impact of more complex lesions on the long-term patency included both patients with intermittent claudication as well as those with CLTI [7,29,30,31].

The deficient long-term efficacy of endovascular treatment of the SFA could also be related to the haemostatic potential, in which the central event is thrombin generation. The studies by Elad et al. [32], and Liew et al. [33] provide some insight into the relationship between thrombin generation and cardiovascular disease progression. Elad et al. found that inhibition of thrombin formation correlates with the progression of coronary disease, while Liew et al. found inhibited thrombin generation in patients with stable PAD compared to a healthy population. We detected only a prolongation of the lag phase in thrombin generation in our patients with restenosis, possibly suggesting a shift toward a hypocoagulable state. Overall, the complex relationship between thrombin generation and disease progression demands further research to fully understand the role of thrombin in the development and progression of PAD.

Plasma levels of EV were elevated in patients with acute coronary syndrome or ischaemic stroke, as well as in individuals with risk factors for cardiovascular disease such as smoking, metabolic disease, and arterial hypertension [34,35,36,37]. We compared the effect of plasma EV on thrombin generation in subjects with and without SFA restenosis and found a greater contribution of EV to thrombin generation in cases of restenosis. This might indicate a higher number of prothrombogenic EV in the blood of patients who later developed restenosis; it is important to note that our study did not quantify EV, determine their source cell type, or identify their phospholipid composition. Saenz-Pipaon et al. [38] observed that patients with PAD released more procoagulant EV of platelet origin. They also found a correlation between an EV protein calprotectin and future events such as amputation. The study by Verwer MC et al. [39] found an association between the increased serpin G1 and CD14 plasma EV protein levels and major adverse cardiovascular events in patients with severe PAD, while elevated serpin G1 plasma EV protein levels were also independently associated with major limb adverse events following SFA endarterectomy.

Maga et al. [40] studied the impact of inflammation on the occurrence of SFA restenosis and concluded that a systematic measurement of leukotriene 4 in the urine of patients undergoing PTA might help in distinguishing patients at risk of restenosis. In our study, only CRP was measured among the inflammation markers and its value did not express a significant relationship with the likelihood of restenosis.

Several studies have demonstrated that genetic variations in the PECAM1 gene can affect the progression of atherosclerosis in the coronary and carotid arteries [41,42,43]. Based on their findings, we hypothesised that PECAM1 gene variations could also impact the progression of PAD and the likelihood of SFA restenosis following PTA. However, our study failed to confirm an association between the studied rs668 and rs12953 gene polymorphisms and the probability of SFA restenosis. A meta-analysis investigating the association between the PECAM1 SNPs and the risk of heart attack discovered a higher risk in patients with the GG genotype of the rs1131012 polymorphism, but found no significant differences for the rs668 and rs12953 polymorphisms [44]. Unfortunately, our study did not investigate the rs1131012 polymorphism. Similarly, one of our previous observational studies also failed to establish a correlation between these gene polymorphisms and atherosclerosis, as it did not confirm the association of PECAM1 SNPs with either established PAD or adverse cardiovascular events such as death, heart attack, stroke, or CLTI [45]. A study by Bonta et al. [46] found a link between genetic variations in the NR4A2 gene and the risk of coronary artery restenosis. However, in our study, we found the NR4A2 gene polymorphisms to be equally common in subjects with and without SFA restenosis. In contrast to our findings, Božič et al. [47] found an increased risk of restenosis in patients with NR4A2 haplotypes 2 and 3 one year after PTA of SFA, but their study also included patients with CLTI.

Our study has several limitations. The main limitation is that it was a single-centred analysis of consecutive patients with a modest number of participants. We did not assess collateral circulation in patients who experienced arterial reocclusion, nor did we conduct functional tests to evaluate their physical performance. We have no data on patients’ physical activity after the revascularisation, which could have affected the outcome. Furthermore, we did not measure the quantity of EV or determine their originating cell type, nor did we identify their phospholipid composition. Additionally, we did not examine the patients’ adherence to medication and control their blood pressure and lipid profile on follow-up visits, factors that could potentially influence the occurrence of restenosis.

## 5. Conclusions

The results of our study indicate poor infrapopliteal runoff as a significant risk factor for restenosis. This finding raises the question of the potential benefit of simultaneous revascularisation of significant lesions in below-knee arteries along with SFA revascularisation in patients with intermittent claudication. Further studies are needed to answer this question.

Our study found no correlation between the thrombin generation potential and the risk of SFA restenosis. Laboratory results indicated a possible shift towards hypocoagulability in patients who developed restenosis, with the only statistically significant differences being the prolongation in the lag phase time. On the other hand, the involvement of EV in endogenous thrombin formation was greater in patients with restenosis. Additional research is required to investigate this topic.

We did not find any association between the NR4A2 rs1466408, NR4A2 rs13428968, NR4A2 rs12803, PECAM1 rs668, and PECAM1 rs12953 polymorphisms and the risk of SFA restenosis after PTA.

## Figures and Tables

**Figure 1 jcm-12-06343-f001:**
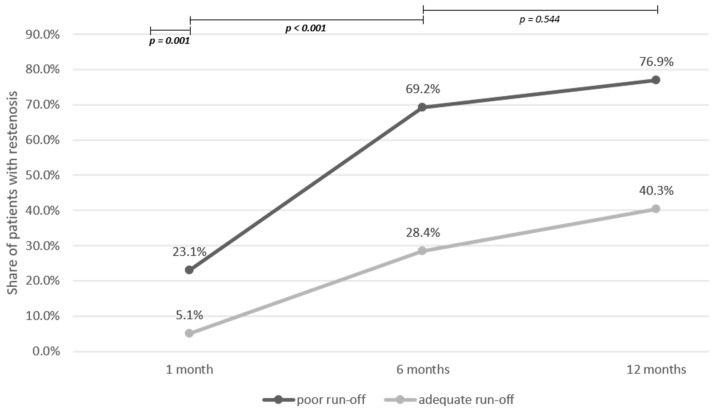
Proportion of 202 patients with superficial femoral artery restenosis 1, 6, and 12 months after the percutaneous transluminal angioplasty in the group of patients with poor infrapopliteal runoff and the group of patients with adequate runoff. Bold values indicate statistical significance at a level of *p* < 0.05.

**Table 1 jcm-12-06343-t001:** Patients’ baseline characteristics and the characteristics of the treated lesions at inclusion in the study.

Patients’ Characteristics (N = 206)	Value
Clinical	
Age (years)	67 ± 9
Male sex	142 (69)
Arterial hypertension	184 (89)
Dyslipidaemia	186 (90)
Diabetes mellitus	78 (38)
Chronic kidney disease (eGFR < 60), N = 196	18 (9)
Smoking (current/abstinence > 1 year)	70/95 (34/46)
Ankle-brachial index before revascularisation	0.64 ± 0.21
Ischaemic heart disease	47 (23)
Prior ischaemic stroke or TIA	41 (20)
Treatment with ACE or ARB	174 (85)
Treatment with statins	183 (89)
Laboratory	
Haemoglobin (g/L), N = 189	142 ± 14
Platelet count (×109/L), N = 189	233 ± 66
eGFR (mL/min), N = 196	87 ± 22
Lesions’ characteristics (N = 206)	
TASC II classification	
A	66 (32.0)
B	103 (50.0)
C	36 (17.5)
D	1 (0.5)
Stenosis/occlusion	108/98 (52/48)
Calcification	77 (37)
Preliminary interventions in the same/different segment	22/49 (11/24)

Data are presented as number and proportion of subjects (N (%)) or as mean ± standard deviation. Estimated glomerular filtration rate (eGFR) was calculated by the MDRD (Modification for Diet and Renal Disease) equation. TIA—transitory ischaemic attack, ACE—angiotensin converting enzyme inhibitor, ARB—angiotensin II receptor blocker, TASC—Trans-Atlantic Inter-Society Consensus [6].

**Table 2 jcm-12-06343-t002:** Comparison of the group of patients with superficial femoral artery restenosis and the group of patients without restenosis.

	Restenosis (N = 91)	No Restenosis (N = 111)	*p*
Clinical Characteristics			
Age (years)	68 ± 9	66 ± 8	0.071
Male gender	59 (64.8)	79 (71.2)	0.336
Arterial hypertension	76 (83.5)	105 (94.6)	**0.010**
Dyslipidaemia	81 (89.0)	103 (92.8)	0.348
Diabetes mellitus	40 (44)	37 (33.3)	0.122
Chronic kidney disease (N = 139)	9 (10.5)	9 (8.6)	0.656
Smoking (active or former)	69 (75.8)	92 (82.9)	0.215
Ischaemic heart disease	23 (25.3)	24 (21.6)	0.541
Prior ischaemic stroke or TIA	23 (25.3)	18 (16.2)	0.111
Former procedures	37 (40.7)	33 (29.7)	0.104
ABI after revascularisation (N = 198)	0.85 ± 0.17	0.93 ± 0.17	**0.001**
Poor infrapopliteal runoff	20 (22.0)	6 (5.4)	**<0.001**
Stent implantation	18 (19.8)	23 (20.7)	0.869
Treatment with ACE or ARB	74 (81.3)	97 (87.4)	0.234
Treatment with statins	80 (87.9)	101 (91.0)	0.476
Dual antiplatelet treatment	34 (37.4)	36 (32.4)	0.464
Characteristics of the treated lesions			
TASC II classification			
A	15 (16.5)	49 (44.1)	**<0.001**
B	53 (58.2)	48 (43.2)	
C	22 (24.2)	14 (12.6)	
D	1 (1.1)	0	
Complete occlusion	50 (54.9)	45 (40.5)	**0.041**
Calcification	32 (35.2)	44 (39.6)	0.514

Data are presented as number and proportion of subjects (N (%)) and as mean ± standard deviation for age and ankle-brachial index (ABI). Bold values indicate statistical significance at a level of *p* < 0.05. TIA—transitory ischaemic attack, ACE—angiotensin converting enzyme inhibitor, ARB—angiotensin II receptor inhibitor, DAPT—dual antiplatelet therapy, TASC—Trans-Atlantic Inter-Society Consensus [6].

**Table 3 jcm-12-06343-t003:** Comparison of laboratory indicators of haemostasis and inflammation between the group of patients with superficial femoral restenosis and the group of patients without restenosis.

	N	Restenosis	No Restenosis	*p*
Coagulation parameters				
D-dimer (μg/L)	85/109	570 (395–835)	540 (395–775)	0.315
PT (rel.)	85/109	0.99 (0.91–1.04)	0.97 (0.91–1.04)	0.959
APTT (s)	85/109	34.4 (31.4–37.5)	34.0 (31.4–38.3)	0.804
Fibrinogen (g/L)	85/109	3.6 (3.1–4.2)	3.5 (3.1–4.0)	0.561
Thrombin generation				
Lag phase (min)	84/108	12 (10–14)	11 (9–13)	**0.042**
Peak thrombin (nM)	84/108	462 (368–562)	493 (396–588)	0.349
Time to peak (min)	84/108	16 (13–18)	15 (12–17)	0.055
Velocity (nM/min)	84/108	135 (96–196)	147 (108–203)	0.213
ETP (nM x min)	84/108	4552 (4101–4951)	4570 (4090–4944)	0.938
EV (%)	84/108	44.7 (36.6–51.7)	37.3 (26.0–46.4)	**<0.001**
EV (nM)	84/108	189 (160–244)	164 (128–229)	**0.006**
Haemostatic potential				
OHP (Abs-sum)	84/105	12.5 (10.1–16.9)	12.6 (10.0–15.1)	0.669
OCP (Abs-sum)	84/105	27.4 (22.1–31.6)	27.0 (22.4–30.1)	0.355
OFP (%)	84/105	53 (44–60)	52 (44–59)	0.648
CRP	85/110	1.0 (0.4–2.7)	1.0 (0.4–2.8)	0.692

Data are shown as the median and range between the first and third quarters. Bold values indicate statistical significance at a level of *p* < 0.05. N—number of samples, PT—prothrombin time, APTT—activated partial thromboplastin time, ETP—endogenous thrombin potential, EV—plasma extracellular vesicles—contribution of EV to thrombin generation, OHP—overall haemostatic potential, OCP—overall coagulation potential, OFP—overall fibrinolytic potential, CRP—C-reactive protein.

**Table 4 jcm-12-06343-t004:** Frequency of single nucleotide polymorphisms of the NR4A2 and PECAM1 genes in subjects with superficial femoral artery restenosis and subjects without restenosis.

Polymorphism	Restenosis N (%)	No Restenosis N (%)	*p*
NR4A2 rs1466408 (N = 154)	67 (100)	87 (100)	
TT	60 (89.6)	76 (87.4)	0.675
TA	7 (10.4)	11 (12.6)	
AA	0	0	
NR4A2 rs13428968 (N = 157)	70 (100)	87 (100)	
TT	51 (72.9)	66 (75.9)	0.766
TC	18 (25.7)	17 (19.5)	
CC	1 (1.4)	4 (4.6)	
NR4A2 rs12803 (N = 160)	72 (100)	88 (100)	
GG	22 (30.6)	31 (35.2)	0.576
GT	35 (48.6)	40 (45.5)	
TT	15 (20.8)	17 (19.3)	
PECAM1 rs668 (N = 141)	65 (100)	76 (100)	
GG	15 (23.1)	24 (31.6)	0.213
GC	28 (43.1)	32 (42.1)	
CC	22 (33.8)	20 (26.3)	
PECAM1 rs12953 (N = 141)	65 (100)	76 (100)	
AA	19 (29.2)	27 (35.5)	0.315
AG	31 (47.7)	36 (47.4)	
GG	15 (23.1)	13 (17.1)	

Data are presented as number and proportion of samples (N (%)).

**Table 5 jcm-12-06343-t005:** Frequency of individual haplotypes of the NR4A2 gene in the group of subjects with superficial femoral artery restenosis and the group of subjects without restenosis.

Haplotype	rs1466408	rs13428968	rs12803	Restenosis (%)	No Restenosis (%)	*p*
1	T	T	G	51	59	0.190
2	T	T	T	29	21	0.809
3	T	C	T	13	14	0.776
4	A	T	T	6	6	0.773

Data are presented as relative frequencies of each haplotype.

**Table 6 jcm-12-06343-t006:** Multivariate analysis of factors associated with restenosis.

Risk Factor.	OR	95-% CI	*p*
Age	1.04	1.00–1.08	0.077
Male sex	1.25	0.59–2.65	0.569
Poor infrapopliteal runoff	5.72	1.73–18.95	**0.004**
TASC II (B vs. A)	3.43	1.51–7.79	**0.003**
TASC II (C vs. A)	9.83	3.34–28.95	**<0.001**
Complete occlusion	1.53	0.76–3.09	0.233
Arterial hypertension	0.21	0.07–0.67	**0.009**
Lag phase in thrombin generation	1.20	1.05–1.36	**0.008**
EV (nm)	1.01	1.00–1.01	**0.001**

Bold values indicate statistical significance at a level of *p* < 0.05. OR—odds ratio, CI—confidence interval, TASC—Trans-Atlantic Inter-Society Consensus [6], EV—contribution of plasma extracellular vesicles to thrombin generation.

## Data Availability

The data presented in this study are available on request from the corresponding author. The data are not publicly available due to ethical considerations.

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
