# Peer review of "Prognostic Factors for Restenosis of Superficial Femoral Artery after Endovascular Treatment"

_jcm, 2023, doi:10.3390/jcm12196343_

Round 1

Reviewer 1 Report

Boc et al. present a fairly well performed analysis of the factors that might prognosticate restenosis of femoral artery after endovascular treatment. Although there are limitations to the study due to having a modest number of participants and a lack of functional characterizations, there are some interesting findings that arise from it. 

I only have one minor comment:

Page 2, Line 71 – Expand TASC II here. 

Author Response

Thank you.

Best regards

Vinko Boc

Reviewer 2 Report

Dear Authors,

You raised an important topic of the recurrence of restenosis after endovascular treatment and tried to find the main causes and factors of this still-not-fully understood phenomenon. There are some major and minor issues that I would like you to address and solve before acceptance.

1) In the Introduction, you mentioned already-known factors for poor endovascular treatment outcomes, but you totally omitted the role of inflammation. There is a publication of Maga P. et al that clearly showed that the increased concentration of postoperative leukotriene E4 not only correlates but also predicts the occurrence of restenosis (10.1016/j.atherosclerosis.2016.04.013). Please refer to it in the Introduction or Discussion section. 

2) You mention in multiple places "femoral artery". Which femoral artery? Common, superficial or deep? This needs to be specified, as it makes a huge difference, especially in the treatment approach. 

3) You use the term "critical limb ischemia" on multiple occasions. This term is outdated. Please change to "chronic limb-threatening ischemia (CLTI)"

4) Line 98-100. What was the location of restenosis in accordance with your definition? Had it to be in the same artery that was endovascularly treated? In exact same spot where PTA was performed? Or does de-novo occured stenosis in different arteries also qualified as restanosis? How was the 50% of narrowing measured - by the lumen diameter in B-mode or based on the changes in blood flow (Vmax)? 

5) Methodology - Was it a prospective observational or retrospective study? This needs to be stated clearly.

6) Table 2 - What are the values in "ABI" row represent? They are clearly not the mean or median ABI values. Are those numbers of improvement/decrease of ABI value? If so, the definition of such changes are needed (the absolute change or some % of change?)

7) Line 149-150 + Fig 1. Is this a primary or secondary patency? Are the patients who underwent reintervention up to 6 months still included in the 12-month analysis? If so, this analysis needs to be recalculated without them (this is the most probable reason for no dependence between poor run-off in 1-year observation, while it was significant in 1 and 6-month timepoints.

8) Line 207-208 The recent publication also shows that in patients undergoing endovascular treatment due to IC the level of vascular inflammation plays an important role in developing symptoms and their affection of patients' life quality

8) Lines 229-241 Also study of Kaczmarczyk et al (10.5603/KP.a2018.0212) is worth mentioning. They showed that blood flow speed (assessed by the time of contrast traveling during angiography) in revascularized BTK arteries negatively correlated with treatment outcomes. The result was caused by the fact that blood flow speed was higher in patients with only one patent artery, while in more healthy patients, the speed was not so high due to the splitting of blood flow between 2 or 3 patent vessels. 

9) Line 266-267 Didn't you assess the ABI during follow-up?

10) Study limitations. I would suggest adding that you also did not study the patients' drug-taking compliance and the lipidogram, which could significantly impact the occurrence of restenosis. 

Kind regards

Some style and grammar changes are needed. I highly suggest to invite native-speaker to check the text. 

Author Response

Thank you.

Best regards

Vinko Boc

Reviewer 3 Report

The article “Prognostic factors for restenosis of femoral artery after endo-2 vascular treatment” has a computer title and whose objective is initially described as studying the effect of infrapopliteal arterial runoff, thrombin generation, overall haemostatic potential (OHP), and single nucleotide polymorphisms (SNPs) of the NR4A2 and PECAM1 genes on the likelihood of restenosis, focus on a pertinent topic that currently remains challenging, and this topic is of interest with characteristics :

Summary

Lines 12 to 15 – the objective is initially described as “…studied the effect of infrapopliteal arterial runoff, thrombin generation, overall haemostatic potential (OHP), and single nucleotide polymorphisms (SNPs) of the NR4A2 and PECAM1 genes on the likelihood of restenosis. ”, not explaining what type of effects they are going to study, positive, negative or both?, leads to the fact that the objective appears described in different ways throughout the work. It is suggested that it be corrected, as the objective must be written in the same way throughout the work, eliminating the subjectivity factor in its reading as best as possible.

Line 19 and 20 – sentence appears in the form of a conclusion, before there is a clear and succinct presentation of the results

Line 21 – reference appears to a variable that is not included in those indicated in the objective. Given that the objective of the work specifies the factors that will be evaluated, it is suggested that others be eliminated.

Keyword

No words were selected, but sets of words. It is suggested to use keywords that appear in the Mesh.

Introduction

There is a lack of bibliographical references. It is suggested that missing references be added.

Materials and Methods

Missing: information about the start and end time of data collection; classification of the type of sampling; reference to the classification table used in the ITB and there is a lack of reference used in the classification of stenoses in echoDoppler. It is suggested that this information be included.

Given that they mention optimization of therapy in some individuals during the study, wouldn't this be a bias in the results? Clarification is suggested.

Does it matter if the exams were carried out by the same observer? Clarification is suggested.

Another important issue is the fact that the exam selected for post-procedure follow-up is different from the pre-procedure exam. This could constitute an important bias compromising some conclusions.

Results

It is suggested to enter the number of observations in the title of each figure.

Discussion

As previously mentioned, the objective must be written in the same way throughout the work.

Reorganization of the discussion is suggested, discussion is suggested in the order in which the results are presented, in order to simplify reading.

Line 205 – conclusions should only appear in the corresponding section.

Line 266 and 267 – As previously mentioned, it is suggested to eliminate the analysis of factors not included in the objective of the work, given that the option was to describe them.

Conclusion

Line 326 to 330 - some conclusions are not supported by the study performed. It is suggested that the paragraph be reformulated.

Author Response

Thank you.

Best regards

Vinko Boc

Round 2

Reviewer 2 Report

Dear Authors,

Thank you for the responses and improvements incorporated into the manuscript. At this point, I do not have any further requests or questions. Congratulations on your work!

Kind regards